# New Characterization of Multi-Drug Resistance of *Streptococcus suis* and Biofilm Formation from Swine in Heilongjiang Province of China

**DOI:** 10.3390/antibiotics12010132

**Published:** 2023-01-10

**Authors:** Chun-Liu Dong, Rui-Xiang Che, Tong Wu, Qian-Wei Qu, Mo Chen, Si-Di Zheng, Xue-Hui Cai, Gang Wang, Yan-Hua Li

**Affiliations:** 1College of Veterinary Medicine, Northeast Agricultural University, Harbin 150038, China; 2Heilongjiang Key Laboratory for Animal Disease Control and Pharmaceutical Development, Harbin 150038, China; 3College of Animal Science and Veterinary Medicine, Heilongjiang Bayi Agricultural University, Daqing 163318, China; 4State Key Laboratory of Veterinary Biotechnology, Harbin Veterinary Research Institute, Chinese Academy of Agricultural Sciences, Harbin 150008, China; 5Department of Basic Veterinary Medicine, College of Veterinary Medicine, Shandong Agricultural University, Taian 271002, China

**Keywords:** *Streptococcus suis*, serotype, multi-drug resistance, biofilm, drug resistance gene

## Abstract

The aim of this study was to investigate the antimicrobial resistance profiles and genotypes of *Streptococcus suis* in Heilongjiang Province, China. A total of 29 *S. suis* were isolated from 332 samples collected from 6 pig farms. The results showed that serotypes 2, 4 and 9 were prevalent, and all the clinical isolates were resistant to at least two antibacterial drugs. The most resisted drugs were macrolides, and the least resisted drugs were fluoroquinolones. Resistant genes *ermB* and *aph (3′)-IIIa* were highly distributed among the isolates, with the detection rates of 79.31% and 75.86%. The formation of biofilm could be observed in all the isolated *S. suis*, among which D-1, LL-1 and LL-3 strains formed stronger biofilm structure than other strains. The results indicate that *S. suis* in Heilongjiang Province presents a multi-drug resistance to commonly used antimicrobial drugs, which was caused by the same target gene, the dissemination of drug resistance genes, and bacterial biofilm.

## 1. Introduction

*Streptococcus suis* (*S. suis*), as an important zoonotic pathogen, can cause several symptoms, such as meningitis, arthritis, pneumonia, endocarditis, and septicaemia in pigs and humans [1]. *S. suis* can be divided into 35 serotypes based on *S. suis* capsular antigens [2], and serotypes 2, 4, 7 and 9 are frequently isolated from both healthy and diseased animals [3,4]. In order to control the *S. suis* infections, massive amounts of antimicrobials are used in the swine industry, which leads to the emergence of *S. suis* drug resistance strains worldwide. In Thailand, 99.3% of *S. suis* isolated from humans and pigs had resistance to at least one antibiotic drug [5]. In Korea, 95.6% of *S. suis* isolated from pigs were resistant to clindamycin, tilmicosin, tylosin, oxytetracycline, and more classes of antimicrobials [3]. In addition, most isolated clinical *S. suis* were resistant to clindamycin, tetracycline, and erythromycin in China from 2017 to 2019 [6]. All of the isolates were resistant to three or more categories of antimicrobial. Therefore, it is essential to monitor antimicrobial susceptibility of *S. suis*.

The clinically isolated *S. suis* often showed cross resistance between macrolides, lincosamides, aminoglycosides, chloramphenicol, β-lactams, fluoroquinolones, and tetracyclines [7,8,9]. Especially, high levels of resistance to tetracycline (TET) and macrolides have been reported in many countries, including North America, Asia, and China [10,11,12]. Moreover, *S. suis* strains resistant to fluoroquinolone [13], lincosamides [14], β-lactams [15], and chloramphenicol [16], and aminoglycosides [3] have also been described in previous studies. Although there have been some reports on the emergence of multi-drug-resistant *S. suis* strains, the molecular mechanisms of resistance remain unclear. Currently, examples for resistance determinants belonging to one of these groups have been described for *S. suis*. For instance, tetracycline resistance in streptococci is mainly due to ribosomal protection genes *tet(M)* and *tet(O)*, and less frequently *tet(Q)*, *tet(T)*, and *tet(W)*, and to efflux genes *tet(K)* and *tet(L)* [17]. The expressions of *ermB*, *ermA*, and *ermTR* [18] are an important mechanism among *S. suis* that confers resistance to macrolides. The macrolide efflux genes (*mefE*, *msrD*, and *mefA*) which are transported by different genetic elements, are responsible for encoding the efflux pump, which promotes the development of resistance to antibiotic drugs in bacterial strains. The resistances of the *mef* genes are often acquired. These genes are located in the transposon or carried by plasmid, and are transmitted between bacteria [19]. Mechanisms of resistance specific to lincosamides relate to the modification of the target by a nucleotidyl-transferase, encoded by the *lnu(B)* or *lnu(C)* genes. The aminoglycoside O-phosphotransferases, encoded by the *aph* genes such as *aph(3′)-IIIa*, mediates resistance to several aminoglycosides, including kanamycin and neomycin [17]. The occurrence of bacterial multi-drug resistance has been of concern, but the exact mechanism is unclear. One of the most important reasons may be that some antibacterial drugs of different classes have the same action sites, including the target gene (e.g., 50S ribosomal subunit) [20,21], and another reason may be the horizontal spread of resistance genes [22]. Under the selective pressure of antimicrobials, resistant bacteria can transfer among animals [23].

Moreover, bacterial biofilm is another important factor responsible for the development of the multi-drug resistance of bacteria [24]. Biofilms are heterogeneous structures which are composed of bacterial cells surrounded by a matrix and attached to solid surfaces. It has been reported that sub-minimum inhibitory concentrations (Sub-MIC) of a variety of antibiotics can induce biofilm formation in a number of phylogenetically diverse Gram-positive and Gram-negative bacteria in vitro [25], as well as induce mutagenesis bacteria of cross-resistance to other antibiotics [24]. Biofilm-associated sessile cells are different from planktonic cells in phenotype and physiology, and they have lower sensitivity to antimicrobial agents due to both tolerance and resistance. Bacterial tolerance refers to the ability of microorganisms to survive brief exposure to high concentrations of antimicrobials [26]. However, tolerant bacteria remain sensitive to antibiotics. Although short exposure to high concentrations of antimicrobials has no significant effect on tolerant bacteria, prolonged exposure to antimicrobials above the minimum inhibitory concentration could lead to the development of resistance [27]. Biofilm can slow down the penetration of antibiotics into the bacteria membrane, change the microenvironment for bacteria to survive, limit bacteria growth, and protect the stress response, all of which will lead to bacterial tolerance [28]. Slow growing or non-growing cells can evade killing by exhibiting low activity of common targets of antibiotics, e.g., RNA polymerase and cell-wall biosynthetic enzymes. Bacteria metabolic adaptations could lead to antibiotic tolerance by decreasing production of electron donors [29]. The presence of an established tolerant mutant in the population could provide more opportunities for the occurrence of rarer resistance mutations. Therefore, the biofilm formation of *S. suis* could lead to the emergence of multi-drug resistant strains, which represents a public health risk to both humans and animals. However, the biofilm formation ability and mechanism of the multi-drug resistant strains of *S. suis* clinically isolated have not been completely elucidated.

In the study, 29 isolates of *S. suis* with multiple drug resistance features were collected from 332 clinical samples of pig farms in Heilongjiang Province of China from 2017 to 2018. Multi-drug resistance related factors were analyzed, including serotype, drug resistant phenotypes, drug resistance genes, and biofilm formation ability. Our findings explain some of the reasons for the development of multi-drug resistance by *S. suis*, providing a theoretical basis to understand, prevent, control, and reduce the occurrence of multiple drug resistance in *S. suis* as well as inhibit their biofilms formation ability.

## 2. Results

### 2.1. Isolation and Identification of S. suis

Isolates were collected from pigs with symptoms of clinical mastitis across different regions in Heilongjiang Province (Figure 1). qPCR detection results showed that *gdh* of *S. suis* were positive. The Gram staining of *S. suis* appeared paired or short-chain. Of the 332 samples tested, 29 (8.7%) isolates were further characterized into various serotypes using *cps* genes (Figure 2). In the Harbin-1 farm, six *S. suis* strains were isolated, in which four isolates belonged to serotype 4, and two isolates belonged to serotype 2. In the Harbin-2 farm, three *S. suis* strains were isolated, in which one isolate belonged to serotype 2, one isolate belonged to serotype 9, and one isolate was unidentified. In the Harbin-3 farm, nine *S. suis* strains were isolated, all of which belonged to serotype 4. In the Suihua farm, one *S. suis* strain was isolated and unidentified. In the Daqing farm, three *S. suis* strains were isolated, in which two isolates belonged to serotype 4, and one isolate was unidentified. In the Qiqihar farm, seven *S. suis* strains were isolated, all of which belonged to serotype 2. Taken together, 10 *S. suis* strains of serotype 2 (34.48%), 15 *S. suis* strains of serotype 4 (51.72%), 1 *S. suis* strain of serotype 9 (3.45%), and 3 unidentified *S. suis* strains (10.35%) were isolated (Appendix A).

### 2.2. Antimicrobial Susceptibility Testing

The definition of multidrug resistance (MDR) is resistance to at least one agent in three or more antimicrobial classes. The MICs distributions of 29 isolates are shown in Figure 3, and the list of the MIC_50_ and MIC_90_ values were included in Appendix A, showing that all the clinical isolates were resistant to macrolides, and the least resisted drugs were fluoroquinolones. According to the CLSI breakpoint, the drug resistance rates of erythromycin, tetracycline, chloramphenicol, florfenicol, penicillin K, gentamicin, ceftiofur, and enrofloxacin were 100%, 96.6%, 89.7%, 86.2%, 86.2%, 82.8%, 82.7%, and 69% respectively. All clinical isolates were resistant to at least two antibacterial drugs (Table 1), and isolates S-2, S-1, AH94-2, B1-2, YB58-2, H1-2, AZ45-1, AZ52-1, and 2-5 showed stronger resistance (Figure 3). In addition, strains of serotype 4 (S-2, S-1, AH94-2, B1-2) showed higher resistance levels than other serotypes.

### 2.3. Detection of Antimicrobial Resistance Genes

Several drug resistance genes were identified, including the *ermA*, *ermB*, and *ermTR* genes, the efflux pump *mefA*, *mefE*, and *msrD* genes of macrolides, the *lnu(B)* gene of lincosamides, the *aph (3′)-IIIa* gene of aminoglycosides, and the *tetO* and *tetM* genes of tetracyclines. The results showed that resistance genes *mefE*, *ermB*, and *aph (3′)-IIIa* were highly distributed among the 29 isolates, with the detection rates of 72.41%, 79.31%, and 75.86%, respectively (Appendix A). The percentages of other resistant genes are presented as follows: *ermA* (55.17%), *ermTR* (58.62%), *mefA* (3.45%), *msrD* (10.34%), *lnu(B)* (55.17%), *tetO* (31.03%), *tetM* (44.83%) (Figure 4). It could be also observed that the isolated strains were 100% resistant to macrolides, and corresponding resistance genes could be amplified from most of the strains.

### 2.4. Biofilm Formation Analysis

Biofilm formation ability was observed in all of the isolated *S. suis* strains. According to the OD values, biofilm formation ability could be divided into three groups: strong, medium, and weak. Among these strains, clinical isolates, such as D-1, LL-1, and LL-3 showed strong ability to form biofilm. Isolates G-3, S-2, S-1, GJ1-2, Y10-2, YF74-2, AH94-2, AZ45-1, K-5, Z-5, YB58-2, DZ001-2, H1-2, AZ52-1, AHD-110-6-2, and AY18-2 showed medium ability to form biofilm. However, LQ-5, H3-1, B1-2, 2-5, B9-1, HB10-1, ZL695-2, YHB13-2, DY12-2, and AH94-4 showed weak ability to form biofilm. When compared to *S. suis* ATCC 7007794 (positive control), isolates D-1, LL-1, and LL-3 also exhibited stronger biofilm formation ability, while AY18-2, AZ45-1, AHD110-6-2, AH94-2, YF74-2, Y10-2, DZ001-2, GJ1-2, S-1, S-2, and G-3 exhibited similar biofilm formation ability. Taken together, isolates D-1, LL-1, and LL-3 of serotype 2 *S. suis* were more capable of forming biofilm than serotype 4 (Figure 5).

## 3. Discussion

*S. suis* is one of the leading causes of infectious diseases in pig farms globally. It is also considered as an emerging zoonotic agent for humans exposed to sick pigs or their by-products, with potential significant impact on public health. So far, the control or treatment of *S. suis* infections depends almost entirely on the use of antimicrobials. In the case of treating *S. suis*, antibiotic resistance is a common phenomenon, and there is an increasing trend of multi-drug resistance. An important approach is to focus on how to mitigate bacterial resistance. In order to promote the rational use of drugs, this study investigated the relationship of drug resistance and biofilm formation using a population of twenty-nine *S. suis* strains isolated from Heilongjiang Province.

*S. suis* serotype 2 is the main serotype causing pig and human infections in China [4]. However, in recent years, *S. suis* serotypes 4 and 9 have been frequently isolated from Chinese pig farms [30], and the isolation rate has gradually increased. Serotype 9 has been frequently isolated from diseased farm animals in intensively-reared commercial pig breeds [31], while serotype 4 has a high isolation rate from healthy pigs in China [32]. In some places in China, *S. suis* serotype 4 has become a dominant serotype [3,33]. Our present data identified at least four serotypes during the experimental period (June 2017 to August 2018) in Heilongjiang Province. This study confirmed that serotypes 2, 4, and 9 were prevalent in Heilongjiang. A total of 10 strains of *S. suis* type 2, 15 strains of *S. suis* type 4, and 1 strain of *S. suis* type 9 were identified; the ratios were 34.48%, 51.72%, and 3.45%, respectively. Obviously, *S. suis* serotype 4 was the main serotype in Heilongjiang during the investigation period. Our study found that strains GJ1-2, S-1, and S-2, isolated from the joints and abdomen of diseased pigs, belong to serotype 4, and exhibit strong multidrug resistance. Simultaneously, they have strong abilities of biofilm formation, suggesting that infections caused by serotype 4 *S. suis* are increasing. Although there has been a study reporting that *S. suis* serotype 2 is by far the most common isolated serotype related to disease [34], the increasing emergence of serotype 4 in infected pigs deserves more attention.

The massive use of antibiotics in the livestock industry contributes to the emergence of antibiotic resistance [7]. In China, the average usage of veterinary antibiotics has reached approximately 6000 tons annually, and most of them are used as feed additives, including tetracyclines and macrolides, among others [35]. In addition, lincosamides, macrolides, aminoglycosides, chloramphenicol, and β-lactams are also widely used in the treatment of *S. suis* [5,36]. In order to know the current drug resistance of *S. suis* in Heilongjiang Province, we tested the drug resistance phenotype and resistance gene of the 29 *S. suis* isolates. The detection results of the MICs of *S. suis* showed that the 29 clinical isolates were resistant to a minimum of two antimicrobial agents and a maximum of eight antimicrobial agents. Resistance to fluoroquinolones was the lowest, and the most resisted drugs were macrolides. Fluoroquinolone was relatively sensitive due to the reduction in the usage of fluoroquinolones in pig farms (a practice recently posed in China), while the massive use of macrolides, lincosamides, and tetracyclines could explain the serious drug resistant phenotype in this study.

The study indicated that all the isolated *S. suis* were 100% resistant to macrolides, which have been widely reported as a commonly used antibiotic in pig farms [37]. Drug resistant bacteria exhibit four drug resistance mechanisms, including target modification, active efflux, mutations in ribosomal L4 and L22, and production of inactivated enzymes [38]. Both *erm* and *mef* are acquired resistance genes. They are located in the transposon or carried by the plasmid, and can be transmitted between bacteria [39]. In a previous study, erythromycin and lincomycin developed cross-resistance due to modification of the same drug target site, which has been confirmed in their clinical isolates [40]. In our study, it was observed that 79.31% of the *ermB* genes detected in tylosin-resistant strains isolated from six farms were also resistant to lincomycin. This may be one of the reasons for the resistance to macrolides and lincomycin. Many drug resistance genes are located on integrative conjugative element (ICE), which can be passed through horizontal gene transfer to a new host cell, thus presenting a serious challenge to the pig industry. Transferable 89K-subtype ICEs, carrying *ermB*- and *tetO* genes, can disseminate tetracycline and macrolide resistance genes in different swine and bovine farms of China [41]. Although not determined in this study, ICEs may lead to the resistance of the strains to macrolides and tetracycline in our research.

It is well known that biofilm formation by pathogens can significantly increase their tolerance to antibiotics [42]. Interestingly, *S. suis* tend to increase biofilm formation once exposed to sub-MICs of commonly used antimicrobials, including amoxicillin, enrofloxacin, oxytetracycline, and lincomycin. However, macrolides could inhibit the formation of *S. suis* biofilm in sub-MIC [43,44]. It is likely that some bacterial cells in biofilm are exposed to antibiotics of sub-MIC levels during antimicrobial chemotherapy, due to falling concentrations by dilution, or diffusion gradients for antibiotics in biofilm. Antibiotics of sub-MICs can provide a selective pressure, under which bacteria tend to mutate and become resistant to other antibiotics [24,42]. In our study, the reason for the stronger biofilm formation ability of LL-1, LL-3, and D-1 could be the long term administration of multi-antimicrobials in the pig farms in Qiqihar, which leads to biofilm formation. It has been reported that growing *S. suis* in the presence of sub-MIC of norfloxacin, doxycycline, and gentamicin is associated with increased biofilm formation ability, which promotes antibiotic tolerance [45]. Antimicrobials at sub-MIC can induce the production of extracellular polymeric substance (EPS) and further promote the resistance of biofilm. In this study, AH94-2 and AH94-4 were the same serotype *S. suis* isolated from the same farm (Harbin-1). Although the MIC of AH94-2 and AH94-4 to lincomycin were at the resistance level, and the gene *lnu(B)* was amplified, they showed different sensitivities to β-lactams. AH94-2 strain is more resistant to β-lactams than strain AH94-4. Meanwhile, isolates LL-3 and LQ-5 were obtained from a farm which used lincomycin, yet LL-3 showed stronger resistance to β-lactam drugs than LQ-5. We found that AH 94-2 and LL-3 exhibited stronger biofilm formation capacity than AH94-4 and LQ-5, which could reduce sensitivity to β-lactam antibiotics. That may explain the reason why AH 94-2 and LL-3 were more resistant than AH 94-4 and LQ-5, respectively. Another study also reported that biofilm-grown *S. suis* cells were much more resistant to penicillin G than planktonic cells [46]. Grenier also reported that the biofilm-grown *S. suis* was significantly more resistant to both penicillin G and ampicillin than planktonic cells [47]. It is possible that the EPS matrix forms a physical protective barrier for bacteria, which can prevent antimicrobials from penetrating into bacteria and ensure their normal growth [48]. In addition, the antibiotic degrading enzymes could be accumulated in the process of antibiotic penetration, which could inactivate the antibiotics and protect the bacteria within the biofilm [49]. Moreover, related resistance genes could not be amplified in macrolides resistant strain 2–5, aminoglycoside resistant strains B9-1, AY18-2, GJ1-2, and LL-1, and tetracycline resistant strains LL-1, G-3, DZ001-2, ZL695-2, Y10-2, AY18-2, and AHD-110-6-2, which indicates that the enhancement of biofilm formation may be related to the sensitivity of antimicrobial drugs. In addition, there is another possibility: that the PCR system was not able to detect all drug resistance genes. Notably, strain LL-1 did not contain resistance genes to tetracycline or aminoglycoside, yet exhibited resistance to both antimicrobials, which might be related to the strong biofilm formation ability. Biofilm grown cells may induce different physiological characteristics than the same isolate grown as colonies on the petri plate, and may have slightly different drug resistance phenotypes. Related studies have revealed that biofilm formation in bacteria could lead to the restriction of drug transport, which is the phase of positively charged antibiotics and a negatively charged EPS matrix [45]. Besides the EPS matrix and peptidoglycan, there are also other cell membrane components that could block the penetration of antibiotics. Bacteria could acquire more time from the delay of antibiotic penetration to achieve adaptive stress response [50]. The multi-speciation and heterogeneity of the bacterial community within the biofilm could cause a high probability of gene mutation, which will change the original gene structure and produce resistance via reducing or preventing binding of antibiotics to the target protein [29]. Interestingly, strain G3 contained resistance genes to aminoglycoside, yet did not exhibit resistance to aminoglycoside. It is possible that biofilm grown *S. suis* develops tolerance to antimicrobials, which enables *S. suis* to survive the MIC. One of the reasons for the development of bacteria tolerance is the slow growth of bacteria after biofilm formation, which causes the bacteria to enter into a growth quiescent period, thus reducing the susceptibility of biofilm to antibiotics [28]. Fridman et al. also reported that *Escherichia coli* cells spontaneously exhibited tolerance after repeated treatments of ampicillin [51]. Biofilm formation could provide bacteria with protection against host defense systems or antimicrobial agents, which leads to multidrug resistance and chronic infection of bacteria. Therefore, the biofilm formation is an important reason for the drug resistance of *S. suis* infections.

In conclusion, 29 clinical *S. suis* strains were isolated, and serotypes 2, 4, and 9 of *S. suis* were identified to be prevalent in Heilongjiang Province in this study. Among the isolated strains, the resistance rate of fluoroquinolones was the lowest, while the resistance rate of macrolides was 100%, indicating the serious situation of multi-drug resistance of *S. suis* in Heilongjiang Province. Resistant genes *ermB* and *aph (3′)-IIIa* were highly distributed among the 29 isolates. The inconsistency of the resistance genotypes and phenotypes of *S. suis* was due to the formation of biofilm, or could have been because the PCR/primer systems used in this study did not detect some drug resistant genes. In addition, this study did not measure drug resistance from biofilm grown isolates, only from colonies on petri plates. The physiological difference in growth conditions could also provide a reason for inconsistency in genotype and phenotype. The study provides us with detailed information on the prevalence and antimicrobial susceptibilities of *S. suis* in Heilongjiang Province from 2017 to 2018, which will enhance our knowledge in understanding how to prevent and control the problem of multi-drug resistance caused by *S. suis* in pig farms.

## 4. Materials and Methods

### 4.1. Samples

332 swabs were collected from 332 pigs of six swine farms in different regions of Heilongjiang Province (one sample/pig), during June 2017 to August 2018. The pigs were housed at the six farms under the same housing, feed, and environmental conditions. The room temperature is 22 °C, the room humidity is 55%, and the pigs were fed three times a day. The majority of the isolates were obtained from the nasal mucosa of asymptomatic pigs and others were obtained from the lungs, joints and abdomens of diseased pigs. Samples collected in Daqing, Qiqihar, and Harbin-3 are from animals that were treated with some antibiotics to cure the disease, while the samples from the other three farms are from animals that did not receive any antibiotic therapy (Appendix A). Of the available farm data, six farms accounted for twenty-nine isolates (Table 2). In the Harbin-1 farm, 40 samples were collected and 6 *S. suis* were isolated. In the Harbin-2 farm, 26 samples were collected and 3 *S. suis* were isolated. In the Harbin-3 farm, 44 samples were collected and 9 *S. suis* were isolated. In the Suihua farm, 4 samples were collected and 1 *S. suis* was isolated. In the Daqing farm, 40 samples were collected and 3 *S. suis* were isolated. In the Qiqihar farm, 178 samples were collected and 7 *S. suis* were isolated. These samples were approved and conducted in accordance with the guidelines of the Animal Welfare and Research Ethics Committee of Northeast Agricultural University. Welfare-related assessments and interventions that were carried out prior to, during, and after the experiment under the strict guidance of the committee.

### 4.2. S. suis Isolation and Serotypes Identification

Swabs were used to take nasal mucosa or fresh tissues from the pigs. Samples were plated onto Sheep Blood Agar (SBA) (Oxoid, Thermo Fisher Scientific, Waltham, MA, USA), and incubated at 37 °C for 24 h to 48 h. Isolates that showed alpha hemolysis on SBA were suspected to be *S. suis*. All suspected *S. suis* strains were further identified by classical biochemical methods, such as Gram staining (Solarbio, Beijing, China). Bacterial genome extraction (Takara, Dalian, China) was performed according to the manufacturer’s instructions. The amplification and sequencing of glutamate dehydrogenase (*gdh*) genes were conducted to further molecularly identify the suspected isolates [52]. The *S. suis* serotype 2 strain ATCC 700794 was used as a positive control for PCR amplifications. In previous research, 33 serotypes of *S. suis* have been identified based on antigenic differences in the capsular polysaccharide. Serotypes 2, 4, 7, and 9 are common epidemic strains; serotypes of the 29 *S. suis* isolates were determined by PCR as previously described [53] (Appendix A).

### 4.3. MIC Testing

All isolates were subjected to antimicrobial susceptibility testing using 18 antimicrobial agents via broth microdilution according to the procedure described by the Clinical Laboratory Standards Institute (CLSI). MIC results were categorized as susceptible, intermediate, and resistant, using the clinical interpretation criteria specified in CLSI performance standard VET01-S3 (CLSI, 2015a). If the interpretation criteria were not present in VET01-S3, CLSI performance standard M100-S25 was used (CLSI, 2015b). *Staphylococcus aureus* ATCC 29213 was used as a control strain according to CLSI VET01-S3. The antimicrobial agents tested were as follows: erythromycin (ERY), tylosin (TYL), lincomycin (LMY), kanamycin (KAN), gentamicin (GEN), oxytetracycline (OTET), tetracycline (TET), chlortetracycline (CTET), chloramphenicol (CHL), florfenicol (FFC), ceftiofur (CEF), cefquinome (CEQ), amoxicillin (AML), penicillin K (PK), ofloxacin (OFX), ciprofloxacin (CIP), enrofloxacin (ENR), and dafloxacin (DAN) (Yongkiettrakul et al. 2019). Since the CLSI did not give the MIC breakpoint values of these antibiotics (tylosin, lincomycin, kanamycin, oxytetracycline, chlortetracycline, cefquinome, amoxicillin, ofloxacin, ciprofloxacin, and dafloxacin), the MIC_50_ and MIC_90_ were used for drug sensitivity statistics [54]. The antibiotic drugs were purchased from China Institute of Veterinary Drugs Control (Beijing, China).

### 4.4. Detection of Antimicrobial Resistance Genes

qPCR was conducted for the identification of resistance genes to macrolides (*ermA*, *ermB*, *mefA*, *mefE*, *ermTR*, and *msrD*), tetracyclines (*tetO* and *tetM*), aminoglycosides (*aph (3′)-IIIa*) and lincosamide (*lnu(B)*) (Appendix A). Bacterial genome extraction (Takara, Dalian, China) was carried out according to the manufacturer’s instructions [5]. The reaction conditions of PCRs were as follows: 94 °C for 30 s, followed by 30 cycles of amplification at 94 °C for 30 s, and 60 °C for 30 s (Takara, Dalian, China).

### 4.5. Biofilm Formation

We used a 96-well plate to examine the formation of bacterial biofilm. The plates for biofilm cultivation should be sterile, flat-bottomed, 96-well polystyrene tissue culture treated microtiter plates with a lid. The *S. suis* strains were subcultured on Todd Hewitt Broth (THB) agar plates under sterile conditions for purification separation. A single colony was selected from the THB agar plates and inoculated in sterile THB liquid test tubes. The incubation was seeded in three wells using 200 μL of each sample. Negative control wells contained broth only. The test for each strain was repeated three times. After incubation at 37 °C for 72 h without shaking, the medium was removed by aspiration and the wells were washed twice with distilled water. The remaining attached bacteria were fixed with 200 µL of 99% methanol (Guoyao Ltd., China) per well, and the plates were emptied after 5 min and left to dry. Then, the plates were stained with 200 µL of 2% crystal violet (Guoyao Ltd., Shanghai, China) per well for another 5 min. The excess stain was rinsed off by placing the plate under running tap water. After the plates were air dried, the dye bound to the adherent cells was resolubilized with 200 µL of 33% (*v*/*v*) glacial acetic acid (Guoyao Ltd., China) per well. The amount of released stain was quantified by measuring the absorbance at 595 nm with a microplate reader (DG5033A, Huadong Ltd., Nanjing, China) [55]. The experiments were performed in triplicate, and the mean values of three measurements were reported.

### 4.6. Statistical Analysis

Statistical analysis of in vitro experiments was performed using GraphPad software. An unpaired t test was performed on qRT-PCR results. Values of *p* < 0.05 were considered significant, and all data were expressed as mean ± SD. *** *p* < 0.001; ** *p* < 0.01; * *p* < 0.05. In order to calculate the ability of biofilm formation accurately, the cut-off value (ODc) was required to be established to separate biofilm producing strains from non-biofilm-producing strains. The cut-off value (ODc) was defined as five standard deviations (SD) above the mean OD of the negative control. ODc = average OD of negative control + (5 × SD of negative control). Final OD value of a tested strain is expressed as average OD value of the strain reduced by ODc value (OD = average OD of a strain-ODc). ODc value is calculated for each microtiter plate separately. If a negative value is obtained, it should be presented as zero, while any positive value indicates biofilm production. Biofilm formation ability was classified according to the values of the ODs obtained: OD ≤ ODc, non-adherent; ODc < OD ≤ 2 × ODc, weak biofilm formation; 2 × ODc < OD ≤ 4 × ODc, moderate biofilm formation; and 4 × ODc < OD, strong biofilm formation [56,57]. Controls included THB medium alone (negative) and culture with *S. suis* ATCC 700794 (positive).

## 5. Conclusions

In this study, 29 clinical *S. suis* strains were isolated, and serotypes 2, 4 and 9 of *S. suis* were identified to be prevalent in Heilongjiang Province. Among the isolated strains, the resistance rate of fluoroquinolones was the lowest, while the resistance rate of macrolides was 100%, indicating the serious situation of multi-drug resistance of *S. suis* in Heilongjiang Province. In conclusion, the study provides us with detailed information on the prevalence and antimicrobial susceptibilities of *S. suis* in Heilongjiang Province from 2017 to 2018, which will enhance our knowledge in understanding how to prevent and control the problem of multi-drug resistance caused by *S. suis* in pig farms.

## Figures and Tables

**Figure 1 antibiotics-12-00132-f001:**
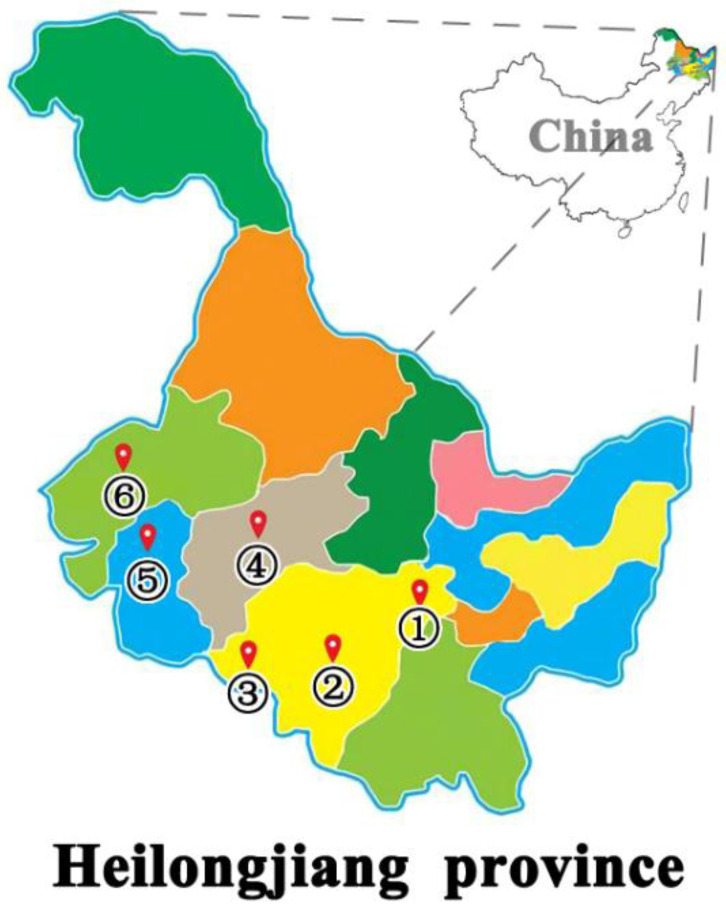
Geographical distribution of the farms where the experimental samples were collected. Numbers 1–6 represent the different farms where samples were collected and the number of samples; 1: Harbin-1 (40), 2: Harbin-2 (26), 3: Harbin-3 (44), 4: Suihua (4), 5: Daqing (40), 6: Qiqihar (178).

**Figure 2 antibiotics-12-00132-f002:**
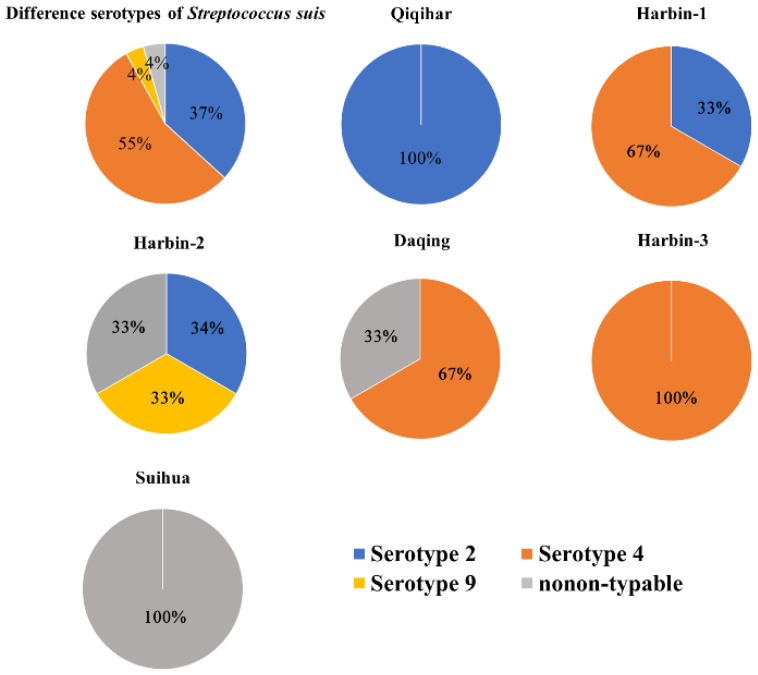
Percentage isolation rate of the different serotypes of *S. suis* isolated from the six farms. *S. suis* 2 were 34.48%, *S. suis* 4 were 51.72%, *S. suis* 9 were 3.45%, and *S. suis* with no serotype were 10.35%. Only one serotype existed in the farms located in Qiqihar, Harbin-3, and Suihua. The farms in Harbin-1 and Daqing had two serotypes. Three serotypes were isolated the farm located in Harbin-2.

**Figure 3 antibiotics-12-00132-f003:**
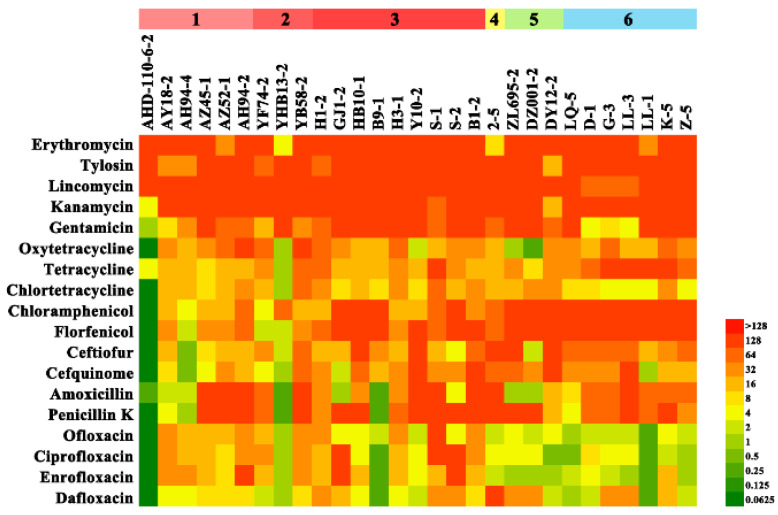
The MIC results of 29 clinical isolates of *S. suis*. The MIC value is represented by a color scale (top right) going from low (green) to high (red).

**Figure 4 antibiotics-12-00132-f004:**
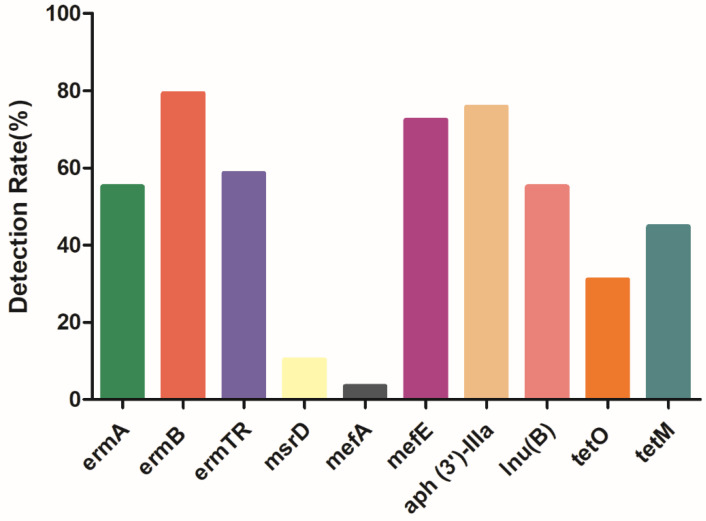
Detection rate of drug resistance genes, *ermA* (55.17%), *ermB* (79.31%), *ermTR* (58.62%), *msrD* (10.34%), *mefA* (3.45%), *mefE* (72.41%), *aph (3′)-IIIa* (75.86%), *lnu(B)* (55.17%), *tetO* (31.03%), *tetM* (44.83%). The resistance genes of *ermA*, *ermB*, *ermTR*, *mefA*, *mefE*, *msrD* belong to macrolides, the *aph (3′)-IIIa* gene belongs to aminoglycosides, the *lnu(B)* gene belongs to lincosamides, the *tetO* and *tetM* genes belong to tetracyclines.

**Figure 5 antibiotics-12-00132-f005:**
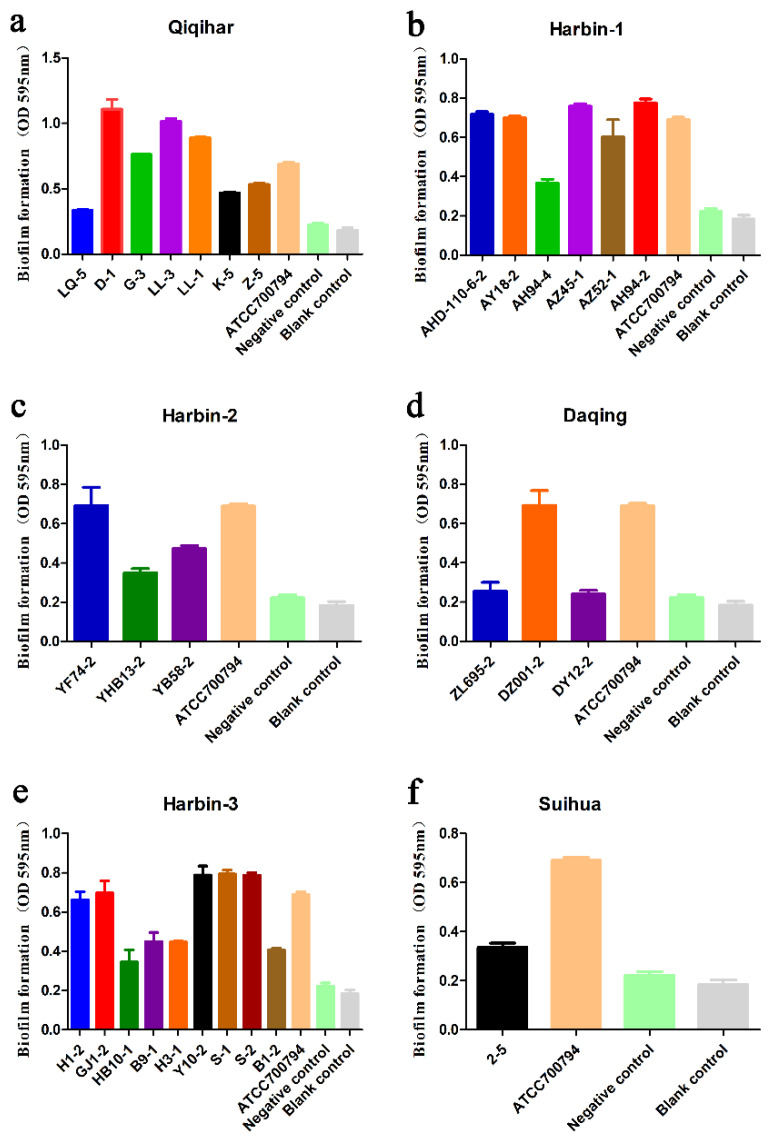
The ability of *S. suis* isolated from the six farms to form biofilm. Data are expressed as means ± standard deviations. *S. suis* ATCC 700794 was used as the positive control, and the negative control was culture with medium only. (**a**) The ability of *S. suis* isolated from Qiqihar pig farm to form biofilm; (**b**) The ability of *S. suis* isolated from Harbin-1 pig farm to form biofilm; (**c**) The ability of *S. suis* isolated from Harbin-2 pig farm to form biofilm; (**d**) The ability of *S. suis* isolated from Daqing pig farm to form biofilm; (**e**) The ability of *S. suis* isolated from Harbin-3 pig farm to form biofilm; (**f**) The ability of *S. suis* isolated from Suihua pig farm to form biofilm.

**Table 1 antibiotics-12-00132-t001:** The antimicrobial resistant types of *S. suis* strains isolated from pigs.

Serial Number	Serotype	Antimicrobial Resistant Types *	No. of Antimicrobials
AHD-110-6-2	4	ERY-TET	2
AY18-2	4	CHL-CEF-ENR-ERY-PK-FFL-TET	7
AH94-4	4	ENR-ERY-GEN-TET	4
AZ45-1	2	CHL-CEF-ENR-ERY-GEN-PK-FFL-TET	8
AZ52-1	2	CHL-CEF-ENR-ERY-GEN-PK-FFL-TET	8
AH94-2	4	CHL-CEF-ENR-ERY-GEN-PK-FFL-TET	8
YF74-2	unidentified	CEF-ENR-ERY-GEN-PK-TET	6
YHB13-2	2	ENR-ERY-GEN	3
YB58-2	9	CHL-CEF-ENR-ERY-GEN-PK-FFL-TET	8
H1-2	4	CHL-CEF-ENR-ERY-GEN-PK-FFL-TET	8
GJ1-2	4	CHL-CEF-ENR-ERY-GEN-PK-FFL-TET	8
HB10-1	4	CHL-CEF-ENR-ERY-GEN-PK-FFL-TET	8
B9-1	4	CHL-CEF-ERY-GEN-FFL-TET	6
H3-1	4	CHL-CEF-ENR-ERY-GEN-PK-FFL-TET	8
Y10-2	4	CHL-CEF-ENR-ERY-GEN-PK-FFL-TET	8
S-1	4	CHL-CEF-ENR-ERY-GEN-PK-FFL-TET	8
S-2	4	CHL-ENR-ERY-GEN-PK-FFL-TET	7
B1-2	4	CHL-CEF-ENR-ERY-GEN-PK-FFL-TET	8
2-5	unidentified	CHL-CEF-ENR-ERY-GEN-PK-FFL-TET	8
ZL695-2	unidentified	CHL-CEF-ERY-GEN-PK-FFL-TET	7
DZ001-2	4	CHL-ERY-GEN-PK-FFL-TET	6
DY12-2	4	CHL-CEF-ERY-GEN-PK-FFL-TET	7
LQ-5	2	CHL-CEF-ENR-ERY-GEN-PK-FFL-TET	8
D-1	2	CHL-CEF-ENR-ERY-PK-FFL-TET	7
G-3	2	CHL-CEF-ENR-ERY-PK-FFL-TET	7
LL-3	2	CHL-CEF-ERY-PK-FFL-TET	6
LL-1	2	CHL-CEF-ERY-GEN-PK-FFL-TET	7
K-5	2	CHL-CEF-ENR-ERY-GEN-PK-FFL-TET	8
Z-5	2	CHL-CEF-ERY-GEN-PK-FFL-TET	7

* CHL, Chloramphenicol; CEF, Ceftiofur; ENR, Enrofloxacin; ERY, Erythromycin; GEN, Gentamicin; PK, Penicillin K; FFL, Florfenicol; TET, Tetracycline.

**Table 2 antibiotics-12-00132-t002:** Number of isolates and sites of regions used in the study.

Isolation Site	Number of Samples	Isolation *S. suis*
Harbin-1	40	6
Harbin-2	26	3
Harbin-3	44	9
Suihua	4	1
Daqing	40	3
Qiqihar	178	7

## Data Availability

Not applicable.

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
