# Peer review of "New Characterization of Multi-Drug Resistance of Streptococcus suis and Biofilm Formation from Swine in Heilongjiang Province of China"

_antibiotics, 2023, doi:10.3390/antibiotics12010132_

Round 1
Reviewer 1 Report
Antibiotics, 2022, entitled „New characterization of multi-drug resistance of Streptococcus suis and biofilm formation from swine in Heilongjiang Province of China” (Dong et al).
Out of 332 samples from 6 pig farms 29 isolates of Streptococcus suis were isolated. The isolates were biochemically and molecular biologically determined and characterized and assigned to the corresponding serotypes. Resistance to numerous antibiotics was determined both phenotypically and genotypically. All parts of the manuscript are well written, clearly arranged and good to understand. Some background information such as frequency occurrence, clinic and existing resistances are clearly described and very valuable in the context.
Some remarks:
Figure 2: in addition to the percentage distribution, the number of isolates and the corresponding serotype could be differentiated. This recognises that distinct serotypes are present in varying numbers, either predominantly or associated with other serotypes on certain farms.
Table 2: More details could be presented here. Resistance to which antibiotics, please list them. Differentiation of serotypes to the multidrug resistance for analyses, which serotype possesses frequently more resistance genes.
Table 3: The designation of the number of isolates is unclear. The number of isolates is high in numbers of S. suis that were found in antibiotic treated farms while less isolates were detected in antibiotic free farms. Please describe the table in more details.
Material and Methods:
S. suis was isolated from 6 farms in the years 2017 and 2018. Are the results from single sampling or multiple samplings. In the case of multiple sampling, the isolate could also have settled in the farm, so that it was repeatedly isolated. Please explain.
The PCR conditions are described. Are these conditions valid for all primers in the form? Was a multiplex-PCR performed?
Biofilm formation: You use a 96-well plate. Is it high binding or low binding, which material? Please specify, because the plates can differ in biofilm formation dependant on surface, material and forming.
Here are some detail remarks:
- Line 29: S. suis italicize
- Line 165-170: resistant genes are genes to resistance, therefore resistance genes
- Line 274: AH94-2 strain is more resistant to AH94-2?
Author Response
Response to Reviewer 1 Comments
Point 1: Figure 2: in addition to the percentage distribution, the number of isolates and the corresponding serotype could be differentiated. This recognises that distinct serotypes are present in varying numbers, either predominantly or associated with other serotypes on certain farms.
Response 1: Thank you very much for your suggestion. The number of isolates and the corresponding serotype have been added in line 119-128.
Point 2: Table 2: More details could be presented here. Resistance to which antibiotics, please list them. Differentiation of serotypes to the multidrug resistance for analyses, which serotype possesses frequently more resistance genes.
Response 2: Thank you for your suggestion. More details have been added in Table 1.
Point 3: Table 3: The designation of the number of isolates is unclear. The number of isolates is high in numbers of S. suis that were found in antibiotic treated farms while less isolates were detected in antibiotic free farms. Please describe the table in more details.
Response 3: Thank you for your suggestion. It is misleading in the designation of the number of isolates. It has been changed to “Number of Samples”. The details were described in line 377-381.
Point 4: Material and Methods:
- suis was isolated from 6 farms in the years 2017 and 2018. Are the results from single sampling or multiple samplings. In the case of multiple sampling, the isolate could also have settled in the farm, so that it was repeatedly isolated. Please explain.
Response 4: Thank you for your suggestion. Single sampling was administered in this study.
Point 5: The PCR conditions are described. Are these conditions valid for all primers in the form? Was a multiplex-PCR performed?
Response 5: Thank you for your suggestion. The annealing temperature of primers were similar, so the conditions were valid for all primers.
Point 6: Biofilm formation: You use a 96-well plate. Is it high binding or low binding, which material? Please specify, because the plates can differ in biofilm formation dependant on surface, material and forming.
Response 6: Thank you for your suggestion. The plates for biofilm cultivation should be sterile flat-bottomed 96-well polystyrene tissue culture treated microtiter plates with a lid. The details of 96-well plate material have been added in line 430-432.
Point 7: Line 29: S. suis italicize
Response 7: Thank you for your suggestion. It has been modified.
Point 8: Line 165-170: resistant genes are genes to resistance, therefore resistance genes
Response 8: Thank you for your suggestion. It has been modified (Line 186).
Point 9: AH94-2 strain is more resistant to AH94-2?
Response 9: Thank you for your suggestion. AH94-2 strain is more resistant to AH94-4. It has been modified (Line 307).
Reviewer 2 Report
Authors;
Manuscript Review
antibiotics-2117646-peer-review-v1
Manuscript Title: New Characterization of Multi-drug Resistance of Streptococ-2 cus suis and Biofilm Formation from Swine in Heilongjiang 3 Province of China
Manuscript Authors: Chun-Liu Dong1,2†, Rui-Xiang Che1†, Tong Wu1, Qian-Wei Qu1, Mo Chen1,Si-Di Zheng1, Xue-hui Cai3, Gang 5 Wang3,4*,Yan-Hua Li1, 2*
Manuscript Summary:
The aim of this study was to investigate the antimicrobial resistance profiles and geno-18 types of Streptococcus suis in Heilongjiang Province, China. A total of 29 S. suis were isolated from 19 332 samples collected from 6 pig farms. The results showed that serotype 2, 4 and 9 were prevalent, 20 and all the clinical isolates were resistant to at least two antibacterial drugs. The most resistant drugs 21 were macrolides, and the least resistant drugs were fluoroquinolones. Resistant genes ermB and aph 22 (3’)-IIIa were highly distributed among the isolates with the detection rates of 79.31% and 75.86%. 23 The formation of biofilm could be observed in all the isolated S. suis among which D-1, LL-1 and 24 LL-3 strains formed stronger biofilm structure than other strains. The results indicate that S. suis in 25 Heilongjiang Province presents a multi-drug resistance to commonly used antimicrobial drugs, 26 which was caused by the same target gene, the dissemination of drug resistance genes, and bacterial 27 biofilm.
Review Summary
Dear authors, thank you for your research paper on a very important topic. I appreciate your hard work and effort. Please read the review and make comments to my questions. Also, there were format issues with two tables. Please read the discussion section suggestions, thank you.
Review:
Abstract.
Introduction.
L36-42. This sentence is too long, please rewrite.
L59. Gene notation are italicized, protein notation are not italicized, please make corrections throughout manuscript. mefE = mefE
L67. “has been concerned” replace with “has been of concern”
L71. Please provide a reference for this statement, thank you.
Methods L323:
4.1 Samples
Please describe in more detail how the pigs were raised or treated.
Were the pigs fed with prophylactic antibiotic in feed? How long, what antibiotic?
Were the pigs born on the farms or transported from another rearing facility?
Table 3.
What does “number of isolates” mean? Were they all S. suis?
For “isolation of S. suis”, was each isolate from only one pig? Were there multiple isolates from one pig?
4.2 S. suis isolation and serotypes identification
L342. “showed alpha hemolysis”
4.4 Detection of antimicrobial resistance genes
Please indicate what type of PCR you used. End point PCR, qPCR, rtPCR?
4.5 Biofilm formation
Please define THB as Todd Hewitt Broth here.
Please define the negative control here.
L381-382. “strain were transplanted” replace with “strains were subcultured”
Results:
2.1 Isolation and identification of S. suis.
L110-111. Did you mean 29 samples or 29 isolates? Was each isolate from one pig or were there multiple isolates from each pig?
Figure 2. Do the farms with only one serotype have any operational practices in common versus the farms with more than one serotype? Any correlation with serotypes?
2.2 Antimicrobial susceptibility testing
Table 1 is very difficult to interpret due to a format problem. Please reformat this table for clarity, thank you.
Figure three . What is the meaning of the numbers above the figure? Are those the farms? Please clarify, thank you.
Were there multiple antibiotics used from each antibiotic class? Are the antibiotics organized by class?
Why is there an extra page in my copy for line 158?
Table 2 also has a format issue. Please reformat table 2 for clarity.
2.3 Detection of antimicrobial resistance genes
L165-166.
It seems that mefE 72.41% should also be included as highly distributed along with ermB and aph (3’)-IIIa
Figure 4. How are the genes grouped in this figure? By class? Please state in legend reason for the order of the genes in the figure.
Discussion.
L206-208. This line maybe considered misleading. “In order to promote the rational use of drugs, we investigated the effects of drug selection pressure on the development of bacterial resistance and biofilm formation, using a population of twenty-nine S. suis strains isolated from Heilongjiang Province.”
Consider replacing with……
“…………………this study investigated the relationship of drug resistance and biofilm formation using a population of twenty-nine S. suis strains isolated from Heilongjiang Province”
L242-244. “Drug 242 resistance bacteria exhibit three drug resistance mechanisms, including target modification, active efflux, and mutations in ribosomal L4 and L22 [37].”
This sentence should also include a phrase concerning lactamase enzymes that inactivate the antibiotic.
L255-256. “The ICEs may lead to the resistance of the 255 strains to macrolides and tetracycline in our study.”
Perhaps replace with….
“Although not determined in this study, ICEs may lead to the resistance of the 255 strains to macrolides and tetracycline in our research.”
L258-261. These lines are confusing here, almost contradictory. Please rephrase this , thank you.
L264. “Antibiotics of Sub-MICs can induce mutagenesis, which confers resistance to other antibiotics [41,44].”
This line was described in other literature but may not be accurate.
Do the antibiotics actually induce mutagenesis or do they provide a selective pressure for random mutation events? Please describe this.
You cited ref 44 here but this is a review, not the original research article. Cite this instead
Jørgensen KM, Wassermann T, Jensen PØ, Hengzuang W, Molin S, Høiby N, Ciofu O (2013) Sublethal ciprofloxacin treatment leads to rapid development of high-level ciprofloxacin resistance during long-term experimental evolution of Pseudomonas aeruginosa. Antimicrob Agents Chemother 57(9):4215–4221. doi:10.1128/AAC.00493-13
L287 & L292. “Moreover, related resistance genes could not be amplified in macrolides resistant strain 2-5, aminoglycoside resistant 288 strains B9-1, AY18-2, GJ1-2 and LL-1, and tetracycline resistant strains LL-1, G-3, DZ001-289 2, ZL695-2, Y10-2, AY18-2 and AHD-110-6-2, which indicates that the enhancement of biofilm formation may be related to the sensitivity of antimicrobial drugs.”
You should add another explanation here that your PCR/primer system was not able to detect all drug resistance genes. Your system could have missed detected of some drug resistant genes and not necessarily replated to biofilm development.
Also, you tested isolates grown from colonies on petri plates for drug resistance phenotype and not grown as a biofilm. Please state this here. Biofilm grown cells may induce different physiological characteristics than the same isolate grown as colonies on the petri plate and may have slightly different drug resistance phenotypes.
L299.Please speculate on the gene mutation mechanisms you mention. Does this occur by horizontal gene transfer? Please comment here.
Also, did you test each isolate for the formation of EPS as a colony?
L318-319. “The inconsistency of the resistance genotypes and phenotypes of S. 318 suis was due to the formation of biofilm.”
Please add………… “or could have been due to the PCR/primer systems used in this study did not detect some drug resistant genes.”
Please considered adding……….
“Also, this study did not measure drug resistance from biofilm grown isolates only from colonies on petri plates. The physiological difference in growth conditions could also provide reason for inconsistency in genotype and phenotype.”
Author Response
Response to Reviewer 2 Comments
Point 1: Introduction. L36-42. This sentence is too long, please rewrite.
Response 1: Thank you for your suggestion. It has been rewrite in line 38-47.
Point 2: L59. Gene notation are italicized, protein notation are not italicized, please make corrections throughout manuscript. mefE = mefE
Response 2: Thank you for your suggestion. It has been modified in line 66.
Point 3: L67. “has been concerned” replace with “has been of concern”
Response 3: Thank you for your suggestion. It has been replaced in line 74.
Point 4: L71. Please provide a reference for this statement, thank you.
Response 4: Thank you for your suggestion. The reference have been added in line 79.
Point 5: Methods L323: 4.1 Samples Please describe in more detail how the pigs were raised or treated.
Response 5: Thank you for your suggestion. The details of pig raising have been added in line 366-369.
Point 6: Were the pigs fed with prophylactic antibiotic in feed? How long, what antibiotic?
Response 6: Thank you for your suggestion. The farms did not use prophylactic antibiotic in feed. The Harbin-3 and Daqing farm only used tylosin to treat the infected pigs. The Qiqihar farm used tylosin, lincomycin, gentamicin, sulfonamide to treat the infected pigs.
Point 7: Were the pigs born on the farms or transported from another rearing facility?
Response 7: Thank you for your suggestion. The pigs were born on the farms.
Point 8: Table 3. What does “number of isolates” mean? Were they all S. suis?
Response 8: Thank you for your suggestion. It is misleading in the “number of isolates”. It has been changed to “Number of Samples”.
Point 9: For “isolation of S. suis”, was each isolate from only one pig? Were there multiple isolates from one pig?
Response 9: Thank you for your suggestion. Only one sample is taken from each pig. Each isolate of S. suis was from only one pig.
Point 10: 4.2 S. suis isolation and serotypes identification.
L342. “showed alpha hemolysis”
Response 10: Thank you for your suggestion. It has been changed in line 391.
Point 11: 4.4 Detection of antimicrobial resistance genes
Please indicate what type of PCR you used. End point PCR, qPCR, rtPCR?
Response 11: Thank you for your suggestion. The qPCR method was used, and it has been modified in line 422.
Point 12: 4.5 Biofilm formation
Please define THB as Todd Hewitt Broth here.
Response 12: Thank you for your suggestion. It has been added in line 433.
Point 13: Please define the negative control here.
Response 13: Thank you for your suggestion. The negative control was added in line 436.
Point 14: L381-382. “strain were transplanted” replace with “strains were subcultured”
Response 14: Thank you for your suggestion. It has been replaced in line 432.
Point 15: Results:
2.1 Isolation and identification of S. suis.
L110-111. Did you mean 29 samples or 29 isolates? Was each isolate from one pig or were there multiple isolates from each pig?
Response 15: Thank you for your suggestion. Sorry for the misleading, it has been modified in line 118. Each isolate was from one pig.
Point 16: Figure 2. Do the farms with only one serotype have any operational practices in common versus the farms with more than one serotype? Any correlation with serotypes?
Response 16: Thank you for your suggestion. One serotype was isolated from farms that administered antimicrobials, while multiple serotypes were isolated from farms that did not use antimicrobials. It is possible that the use of antimicrobials lead to the single serotype.
Point 17: 2.2 Antimicrobial susceptibility testing
Table 1 is very difficult to interpret due to a format problem. Please reformat this table for clarity, thank you.
Response 17: Thank you for your suggestion. The table 1 has been reformatted and moved to supplementary materials.
Point 18: Figure three . What is the meaning of the numbers above the figure? Are those the farms? Please clarify, thank you.
Response 18: Thank you for your suggestion. The numbers represent different farms. No.1 refers to Harbin-1 farm, No.2 refers to Harbin-2 farm, No.3 refers to Harbin-3 farm, No.4 refers to Suihua farm, No.5 refers to Daqing farm, No.6 refers to Qiqihar farm.
Point 19: Were there multiple antibiotics used from each antibiotic class? Are the antibiotics organized by class?
Response 19: Thank you for your suggestion. Multiple antibiotics were used from each antibiotic class. The antibiotics were organized by class (Data shown in table S2).
Point 20: Why is there an extra page in my copy for line 158?
Response 20: Thank you for your suggestion. It has been modified.
Point 21: Table 2 also has a format issue. Please reformat table 2 for clarity.
Response 21: Thank you for your suggestion. The table 2 has been reformatted.
Point 22: 2.3 Detection of antimicrobial resistance genes
L165-166. It seems that mefE 72.41% should also be included as highly distributed along with ermB and aph (3’)-IIIa
Response 22: Thank you for your suggestion. It has been added in line 186-187.
Point 23: Figure 4. How are the genes grouped in this figure? By class? Please state in legend reason for the order of the genes in the figure.
Response 23: Thank you for your suggestion. The genes were grouped by class. The resistance genes of ermA, ermB, ermTR, mefA, mefE, msrD belong to macrolides, the aph (3’)-IIIa gene belongs to aminoglycosides, the lnu(B) gene belongs to lincosamides, the tetO and tetM genes belong to tetracyclines. It has been added in legend.
Point 24: Discussion.
L206-208. This line maybe considered misleading. “In order to promote the rational use of drugs, we investigated the effects of drug selection pressure on the development of bacterial resistance and biofilm formation, using a population of twenty-nine S. suis strains isolated from Heilongjiang Province.”
Consider replacing with……“…………………this study investigated the relationship of drug resistance and biofilm formation using a population of twenty-nine S. suis strains isolated from Heilongjiang Province”
Response 24: Thank you very much for your suggestion. It has been replaced in line 229-231.
Point 25: L242-244. “Drug 242 resistance bacteria exhibit three drug resistance mechanisms, including target modification, active efflux, and mutations in ribosomal L4 and L22 [37].”
This sentence should also include a phrase concerning lactamase enzymes that inactivate the antibiotic.
Response 25: Thank you very much for your suggestion. It has been added in line 267-270.
Point 26: L255-256. “The ICEs may lead to the resistance of the 255 strains to macrolides and tetracycline in our study.”
Perhaps replace with….
“Although not determined in this study, ICEs may lead to the resistance of the 255 strains to macrolides and tetracycline in our research.”
Response 26: Thank you very much for your suggestion. It has been replaced in line 281-282.
Point 27: L258-261. These lines are confusing here, almost contradictory. Please rephrase this , thank you.
Response 27: Thank you very much for your suggestion. It has been rephrased in line 285-288.
Point 28: L264. “Antibiotics of Sub-MICs can induce mutagenesis, which confers resistance to other antibiotics [41,44].”
This line was described in other literature but may not be accurate.
Do the antibiotics actually induce mutagenesis or do they provide a selective pressure for random mutation events? Please describe this.
Response 28: Thank you very much for your suggestion. Antibiotics of Sub-MICs can provide a selective pressure, under which bacteria tend to mutate and become resistant to other antibiotics. It has been modified in line 294-295.
Point 29: You cited ref 44 here but this is a review, not the original research article. Cite this instead
Jørgensen KM, Wassermann T, Jensen PØ, Hengzuang W, Molin S, Høiby N, Ciofu O (2013) Sublethal ciprofloxacin treatment leads to rapid development of high-level ciprofloxacin resistance during long-term experimental evolution of Pseudomonas aeruginosa. Antimicrob Agents Chemother 57(9):4215–4221. doi:10.1128/AAC.00493-13
Response 29: Thank you very much for your suggestion. The ref 44 has been replaced.
Point 30: L287 & L292. “Moreover, related resistance genes could not be amplified in macrolides resistant strain 2-5, aminoglycoside resistant 288 strains B9-1, AY18-2, GJ1-2 and LL-1, and tetracycline resistant strains LL-1, G-3, DZ001-289 2, ZL695-2, Y10-2, AY18-2 and AHD-110-6-2, which indicates that the enhancement of biofilm formation may be related to the sensitivity of antimicrobial drugs.”
You should add another explanation here that your PCR/primer system was not able to detect all drug resistance genes. Your system could have missed detected of some drug resistant genes and not necessarily replated to biofilm development.
Response 30: Thank you very much for your suggestion. Another explanation has been added in line 323-324.
Point 31: Also, you tested isolates grown from colonies on petri plates for drug resistance phenotype and not grown as a biofilm. Please state this here. Biofilm grown cells may induce different physiological characteristics than the same isolate grown as colonies on the petri plate and may have slightly different drug resistance phenotypes.
Response 31: Thank you very much for your suggestion. The explanation has been added in line 326-328.
Point 32: L299. Please speculate on the gene mutation mechanisms you mention. Does this occur by horizontal gene transfer? Please comment here.
Response 32: Thank you very much for your suggestion. The emergence of mutations in nucleic acids is one of the major factors underlying evolution, providing the working material for natural selection. Mechanisms of horizontal gene spread among bacterial strains or species are often considered to be the main mediators of antibiotic resistance.
Point 33: Also, did you test each isolate for the formation of EPS as a colony?
Response 33: Thank you very much for your suggestion. The test was not performed, we will do it in our further study.
Point 34: L318-319. “The inconsistency of the resistance genotypes and phenotypes of S. suis was due to the formation of biofilm.”
Please add………… “or could have been due to the PCR/primer systems used in this study did not detect some drug resistant genes.”
Response 34: Thank you very much for your suggestion. The explanation has been added in line 354-355.
Point 35: Please considered adding……….“Also, this study did not measure drug resistance from biofilm grown isolates only from colonies on petri plates. The physiological difference in growth conditions could also provide reason for inconsistency in genotype and phenotype.”
Response 35: Thank you very much for your suggestion. The explanation has been added in line 355-358.
Reviewer 3 Report
Up to date data regarding Streptococcus suis drug resistance and biofilm formation isolated from swine in China is extremely important, not only due to economic losses to the swine industry, but also because in some Asian countries, it is considered a major public health concern for the general population as well. I found this topic of scientific interest, since it also alerts for the role of swine might have on disseminating antimicrobial resistance. Furthermore, this kind of surveillance is important to develop strategies to address the drug resistance problem. The manuscript is well written and the results will contribute to the current knowledge in the field.
Introduction
Line 36: ; however
Results
The authors didn’t write in the text what they considered to decide that an isolate is multidrug resistant or not. The definition of multidrug resistance (MDR) is resistance to at least one agent in three or more antimicrobial classes. For example, if the isolate is resistant to Ceftiofur, Cefquinome, Amoxicillin, Penicillin K (all belonging to B-lactam group), and also resistant to Ofloxacin, ciprofloxacin, Enrofloxacin and Dafloxacin (all belonging to Fluoroquinolone group), then the isolate is not multidrug resistant, since it was resistant to drugs belonging only two different categories. If the authors didn’t use this criteria than they must rewrite the results according to this definition.
Table 1 is not necessary in the text; it is better to be in the supplementary material.
Add a table (that could replace Table 1) with the 29 isolates susceptibility result, showing the susceptibility of each isolate to all the drugs tested.
Table 2 can be excluded, because in the last column of new table (29 isolates susceptibility result), the authors can add the number of drugs that each isolate was resistant to.
Line 111: add the percentage after the number 29
Lines 112 - 113: add the number of isolates of each serotype before the percentage
Materials and Methods
Lines 329-331: Change the sentence as following “Samples collected in Daqing, Qiqihar and Harbin-3 are from animals that were treated with some antibiotics to cure the disease, while the samples from the other three farms are from animals that did not receive any antibiotic therapy”
Author Response
Response to Reviewer 3 Comments
Point 1: Introduction Line 36: ; however
Response 1: Thank you very much for your suggestion. The sentence has been rewritten in line38-41.
Point 2: Results The authors didn’t write in the text what they considered to decide that an isolate is multidrug resistant or not. The definition of multidrug resistance (MDR) is resistance to at least one agent in three or more antimicrobial classes. For example, if the isolate is resistant to Ceftiofur, Cefquinome, Amoxicillin, Penicillin K (all belonging to B-lactam group), and also resistant to Ofloxacin, ciprofloxacin, Enrofloxacin and Dafloxacin (all belonging to Fluoroquinolone group), then the isolate is not multidrug resistant, since it was resistant to drugs belonging only two different categories. If the authors didn’t use this criteria than they must rewrite the results according to this definition.
Response 2: Thank you very much for your suggestion. We did use this criteria, and the definition of multidrug resistance (MDR) has been added in line153-154.
Point 3: Table 1 is not necessary in the text; it is better to be in the supplementary material.
Response 3: Thank you very much for your suggestion. The table 1 has been moved to supplementary material.
Point 4: Add a table (that could replace Table 1) with the 29 isolates susceptibility result, showing the susceptibility of each isolate to all the drugs tested.
Response 4: Thank you very much for your suggestion. The new table has been added.
Point 5: Table 2 can be excluded, because in the last column of new table (29 isolates susceptibility result), the authors can add the number of drugs that each isolate was resistant to.
Response 5: Thank you very much for your suggestion. The table 2 has been excluded.
Point 6: Line 111: add the percentage after the number 29
Response 6: Thank you very much for your suggestion. The percentage has been added in line 118.
Point 7: Lines 112 - 113: add the number of isolates of each serotype before the percentage
Response 7: Thank you very much for your suggestion. The number of isolates of each serotype has been added in line 126-128.
Point 8: Materials and Methods Lines 329-331: Change the sentence as following “Samples collected in Daqing, Qiqihar and Harbin-3 are from animals that were treated with some antibiotics to cure the disease, while the samples from the other three farms are from animals that did not receive any antibiotic therapy”
Response 8: Thank you very much for your suggestion. The sentence has been changed in line 370-373.